# OPTUNE: EFFICIENT ONLINE PREFERENCE TUNING

## ABSTRACT

Reinforcement learning with human feedback (RLHF) is critical for aligning Large Language Models (LLMs) with human preference. Compared to the widely studied offline version of RLHF, *e.g.* direct preference optimization (DPO), recent works have shown that the online variants achieve even better alignment. However, online alignment requires on-the-fly generation of new training data, which is costly, hard to parallelize, and suffers from varying quality and utility. In this paper, we propose a more efficient data exploration strategy for online preference tuning (OPTUNE), which does not rely on human-curated or pre-collected teacher responses but dynamically samples informative responses for on-policy preference alignment. During data generation, OPTUNE only selects prompts whose (re)generated responses can potentially provide more informative and higher-quality training signals than the existing responses. In the training objective, OPTUNE reweights each generated response (pair) by its utility in improving the alignment so that learning can be focused on the most helpful samples. Throughout our evaluations, OPTUNE'd LLMs maintain the instruction-following benefits provided by standard preference tuning whilst enjoying 1.27-1.56x faster training speed due to the efficient data exploration strategy.

## 1 INTRODUCTION

Reinforcement Learning from Human Feedback (RLHF) has emerged as an effective method for training large language models (LLMs) to generate responses that are more aligned with human preferences (Ziegler et al., 2019b; Ouyang et al., 2022a), and has underpinned the successes of systems like ChatGPT and the Gemini models. Offline preference tuning (PT) techniques such as DPO (Rafailov et al., 2023), IPO (Azar et al., 2024b), and KTO (Ethayarajh et al., 2024) are also viable solutions for utilizing the human preference dataset to enhance the alignment qualities of of LLMs but these techniques require large volumes of annotated response data. Its counterpart, *online* PT, exhibits promising potential but demands continuous sampling of new responses from the LLM policy during iterative training which is an expensive operation in its own right. Considering online DPO training as an example, we can break the overall process down into four steps: (1) Reward model (RM) training. (2) Sampling responses from the trained policy (LLM). (3) Evaluate responses by the rewards from RM. (4) Preference Tuning (PT) on the reward-labeled responses. Given the time-consuming and resource-intensive nature of these steps, our goal in this work is to study methods for expediting the entire training cycle without compromising the quality of the trained models, thereby enhancing the practical feasibility and effectiveness of online DPO.

Based on our analysis, as reported in Table 1, it is evident that generating responses and training the policy model are the most time-consuming steps of online DPO training. Can we naïvely reduce the number of responses being generated? Unfortunately, in preliminary experiments, we find that randomly selecting half of the generated responses for reuse during iterative training results in a significant degradation in

Table 1: Time percentage for each procedure in online DPO. The batch size of generation and training have been optimized for GPUs to ensure good parallelism. We set the max response length of both generation and training to 512.

|  | Generation | Rewarding | Training |
|---|---|---|---|
| Time | 71.8% | 0.1% | 28.1% |

instruction-following performance compared to that of policies trained in a fully online setting. This leads to another question: *Can we maintain the performance of online PT while adhering to a fixed generation budget?*

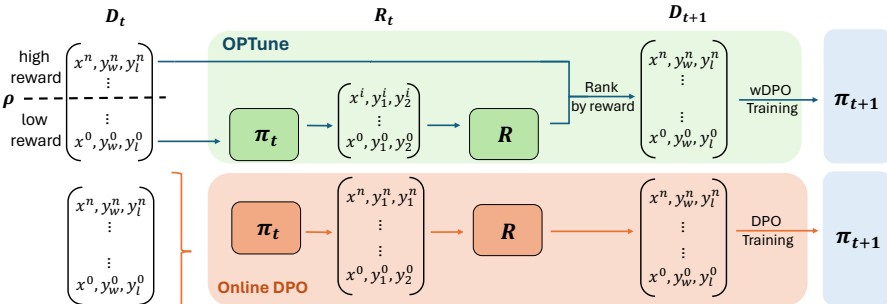

Figure 1: The pipeline of our OPTUNE: it only explores the low-reward examples and reuses the high-quality examples, which improves the generation efficiency of the iterative online PT. We also exploit the weighted DPO to enhance the training efficiency by focusing on the high-utility samples. $\pi_t$: the policy in iter $t$. $R$: the reward model. $\rho$: the prompt selection ratio for re-generations.

First, to reduce the generation cost without compromising instruction-following capabilities or alignment quality, we propose to only re-generate and update the lowest-rewarded responses produced under the latest LLM policy. We posit that the policy's behavior on these specific prompts can likely be improved further than in scenarios where its responses are already high quality potentially leading to greater improvements in overall reward at each step. Thus, we generate new responses for those selected prompts and mix them with the existing high-rewarded responses to constitute the full training set. By implementing the reward-based selection strategy, we address the dual goals of reducing the computational cost of response generation in online DPO while retaining the instruction-following capability, which leads to more data-efficient online RLHF.

Second, we investigate the utility of response pairs in online DPO and propose a weighted DPO (wDPO) objective that focuses learning on preference pairs that may contribute the most to the online alignment process. This is motivated by the simple observation that in the original DPO loss formulation, the positive-negative labels are a binary quantization of their scalar rewards and thus cannot explicitly reflect their reward gap. The reward gap measures the utility of response pairs in DPO training because comparing the preferred and rejected responses with a larger reward gap reveals more clues for improving the alignment. By directly assigning larger weights to these samples, in each round online wDPO concentrates learning on the high-utility samples yielding improved learning efficiency.

We conduct comprehensive experiments to evaluate the OPTUNE-trained LLM policies, incorporating instruction-following evaluations, multiple benchmarks, and human studies. Specifically, we select LIMA (Zhou et al., 2023) and AlpacaEval (Li et al., 2023b) test sets as free-form instruction evaluations and conduct pair-wise comparisons by employing GPT-4 as the judge. Given the potential for biases from the judge to confound model-based evaluations, human studies and benchmark evaluations such as MMLU (Hendrycks et al., 2020b), GSM8k (Cobbe et al., 2021a), and TruthfulQA (Lin et al., 2021) are also included. Through our experiments we demonstrate that OPTUNE trains better LLMs than baselines whilst enjoying 1.27-1.56x training speedup due to its efficient data-exploration strategy.

To sum up, OPTUNE is the first efficient data generation algorithm for online RLHF. By selectively regenerating only the lowest-rewarded responses and using a weighted DPO objective that emphasizes pairs with larger reward gaps, OPTUNE significantly enhances both the generation and training efficiency of the RLHF pipeline, thereby paving the way for a promising future in which preference-aligned LLMs can be developed in a resource-efficient manner.

## 2 PRELIMINARIES

The prevalent RLHF pipeline was proposed by Ziegler et al. (2019a) and adopted by subsequent works including (Stiennon et al., 2020; Nakano et al., 2021; Ouyang et al., 2022b; Bai et al., 2022). The standard method comprises three stages: (1) Supervised Fine-Tuning (SFT) on human-annotated/machine-generated responses; (2) reward model training on preference data; and (3) Reinforcement Learning based on the SFT checkpoint and feedback received from the RM.

**Reward Model Training** Following (Ouyang et al., 2022a; Touvron et al., 2023), we utilize the Bradley-Terry model (Bradley & Terry, 1952) in RM training procedure, which provides a probabilistic framework for predicting preferences based on pairwise comparisons. The goal is to learn a set of parameters $\boldsymbol{\theta}$ that best explains the observed preferences between pairs of possible responses. Specifically, the loss function is given by:

$$\mathcal{L}(\theta) = -\mathbb{E}_{(x,y_w,y_l)\sim\mathcal{D}} \left[ \log \sigma \left( r_{\boldsymbol{\theta}}(x, y_w) - r_{\boldsymbol{\theta}}(x, y_l) \right) \right], \tag{1}$$

where $\sigma(\cdot)$ is the sigmoid function; $r_{\boldsymbol{\theta}}(x, y)$ is the scalar reward from the RM; $y_w$ and $y_l$ denotes chosen and rejected responses, respectively. This loss function represents the negative log-likelihood of the model preferring the chosen response $y_w$ over the rejected response $y_l$ under the Bradley-Terry model.

**RL finetuning** The reinforcement learning stage (Bai et al., 2022; Gao et al., 2022) does not require predefined responses. It further fine-tunes the SFT model $\pi_{\text{SFT}}(y|x) = p(y|x; \theta^{\text{SFT}})$ to maximize the reward $r(x, y)$ under a KL regularization to prevent the model from deviating too far from the SFT model:

$$\underset{\theta}{\text{maximize}} \ \mathbb{E}_{x\sim\mathcal{D}_p} \left[ \mathbb{E}_{y\sim\pi_\theta(y|x)} \left[ r(x, y) \right] - \alpha\mathbb{D}_{KL} \left[ \pi_\theta(y|x) | \pi_{\text{SFT}}(y|x) \right] \right], \tag{2}$$

where $\pi_\theta(y|x) = p(y|x; \theta)$; $\alpha > 0$ is a constant to control the regularization strength; $\mathcal{D}_p$ denotes the prompt set used for sampling the response $y \sim \pi_\theta(y|x)$ from the trained policy and construct pair $(x, y)$ for RL training. Note the KL term here is defined on the conditional distribution $p(y|x; \theta)$ as $\mathbb{D}_{\text{KL}} \left[ \pi_\theta(y|x) | \pi_{\text{sft}}(y|x) \right] = \mathbb{E}_{y\sim p(y|x;\theta)} \left[ \log \frac{\pi_\theta(y|x)}{\pi_{\text{sft}}(y|x)} \right]$.

**DPO** One representative method for preference optimization is DPO (Rafailov et al., 2023). It follows Ziebart et al. (2008) and starts with a closed-form solution for Eq. (2):

$$\pi_r(y \mid x) = \frac{1}{Z(x)} \pi_{\text{ref}}(y \mid x) \exp\left( \frac{1}{\beta} r(x, y) \right), \tag{3}$$

where $Z(x)$ is the partition function: $Z(x) = \sum_y \pi_{\text{ref}}(y \mid x) \exp\left( \frac{1}{\beta} r(x, y) \right)$. Then they rearrange the Eq. (3) and express the reward as a function of the policy:

$$r(x, y) = \frac{1}{\beta_1} \left( \log(Z(x)) + \log\left( \frac{\pi_{t+1}(y|x)}{\pi_t(y|x)} \right) \right), \tag{4}$$

where $\pi_t$ and $\pi_{t+1}$ are the policies on the iteration $t$ and $t+1$, respectively. It aims to optimize an implicit reward function as a binary classification loss:

$$\mathcal{L}_{DPO}(\pi_{t+1}; \pi_t) = -\mathbb{E}_{(x,y_u,y_l)\sim\mathcal{D}} \left[ \log \sigma \left( \beta_1 \log \frac{\pi_{t+1}(y_w|x)}{\pi_t(y_w|x)} - \beta_1 \log \frac{\pi_{t+1}(y_l|x)}{\pi_t(y_l|x)} \right) \right]. \tag{5}$$

While in the standard offline DPO setting (Rafailov et al., 2023) the preference datasets are collected before training begins, Chen et al. (2024c); Dong et al. (2024) extend DPO to the online setting, by sampling two new responses to each prompt at every iteration. These two responses are passed to the reward model to identify the preferred and dispreferred response, thereby training the policy on continuously updated preference data with each iteration.

## 3 METHOD

In this section, we develop OPTUNE to improve both the **data generation efficiency** and **training efficiency** of online preference alignment. First, to reduce the cost of iterative data re-generation in the online setting, we propose a simple but effective reward-based prompt selection strategy that only updates the responses for prompts with the lowest scoring current responses according the reward model. Then, motivated by the observation that the quantization of scalar rewards to binary labels required by the online DPO objective necessarily leads to information loss, we propose a weighted DPO loss variant that prioritizes the learning of response pairs with a larger reward gap, thereby improving online learning efficiency even further.

## 3.1 Data generation efficiency: Reward-based prompt selection

According to the Eq. (2), the ultimate goal of RL finetuning is to maximize the expected reward for the generated responses. We first investigate whether different prompts contribute differently to the total reward gain at each step. For each iteration of online DPO, we generate the response for $x^i \in \mathcal{P}$ and the reward model returns the reward value $r^i$ of each response. We compute the reward gain from prior iteration, and also provide statistics showing how different prompts contribute to the overall reward gain.

As illustrated in Fig. 2, we divide the prompt set into two subsets based on the reward rankings of their preferred responses: the top-50% and the bottom-50%. We then analyze the percentage of reward gains from each subset. For example, in Iter2, when comparing the reward on each prompt to the Iter1, only 31.4% of the reward gain originates from prompts that generated higher-reward responses in the previous iteration (top-50 subset),

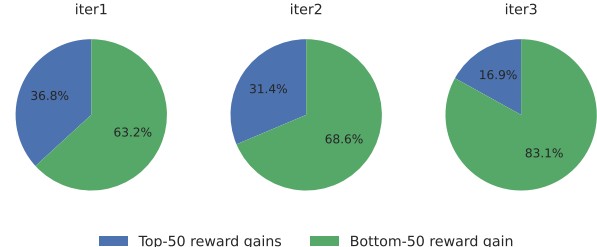

Figure 2: The reward gains brought by two subsets: top-50% ranked prompts and bottom-50% ranked prompts. More gains are achieved from the bottom-50% prompts than the top-50% prompts.

while 68.6% comes from prompts that produced lower-reward responses (bottom-50 subset). That indicates if the response's reward is low in this iteration, the prompt is more likely to produce a high-reward response in the following iteration. Conversely, if the response's reward is high in the current iteration, it is less likely to generate a high-reward response in the next iteration.

Motivated by this observation, we propose a reward-based prompt selection mechanism that prioritizes prompts such that due to their currently low reward, if their responses were to be re-generated and trained on in the next round, the total reward gain of the policy would likely to be larger. Using this selection criteria our algorithm ensures that each training iteration focuses on the most informative examples, thereby improving overall generation efficiency. Algorithm 1 formally defines how OPTune's reward-based prompt selection works.

---

**Algorithm 1** OPTune for Iterative Online DPO

---

1: Initialize policy parameters $\pi_0$; ranked prompt set $\mathcal{P}_t$ and training set $\mathcal{D}_t$ at iteration $t$; Prompt selection ratio $\rho$; generation count $g = 0$;
2: **for** $t = 0$ to $T - 1$ **do**
3:     Clear temporary response storage $\mathcal{R}_t = \{\}$
4:     Calculate the number of prompts to regenerate $N = \lceil \rho \times |\mathcal{P}_t| \rceil$
5:     Set $g = 0$
6:     **while** $g < N$ **do**
7:         Pop the lowest ranking prompt $x^i$ from $\mathcal{P}_t$
8:         Sample two responses $y_1^i$ and $y_2^i$ for $x^i$ using $\pi_t$
9:         Store responses: $\mathcal{R}_t \leftarrow \mathcal{R}_t \cup \{(x^i, y_1^i), (x^i, y_2^i)\}$
10:       Increment the generation count $g = g + 1$
11:     **end while**
12:     **for** each $x^i \in \mathcal{P}_t$ **do**
13:         **if** $(x^i, y_1^i), (x^i, y_2^i) \in \mathcal{R}_t$ **then**
14:            Use the new responses from $\mathcal{R}_t$ for $x^i$
15:         **else**
16:            Use the previous responses from $\mathcal{D}_t$ for $x^i$
17:         **end if**
18:     **end for**
19:     Compute rewards $r_1^i$ and $r_2^i$ for each $(x^i, y_1^i), (x^i, y_2^i) \in \mathcal{R}_t$
20:     Construct the training set $\mathcal{D}_t = \{(x^i, y_w^i), (x^i, y_l^i) \mid x^i \in \mathcal{P}_t\}$
21:     Rank the prompts in $\mathcal{P}_t$ according to rewards to obtain $\mathcal{P}_{t+1}$
22:     Compute the wDPO (or DPO) loss and update the policy parameters $\pi_t$ to obtain $\pi_{t+1}$
23: **end for**

---

## 3.2 TRAINING EFFICIENCY: WEIGHTED DPO LOSS

To improve training efficiency, we more closely examine the iterative online DPO algorithm presented in Algorithm 2.

---

**Algorithm 2** Iterative Online DPO

---

1: Initialize policy parameters $\pi_0$ and prompt set $\mathcal{P}$
2: **for** $t = 0, 1, \ldots, T - 1$ **do**
3:      Sample two responses $y_1^i$ and $y_2^i$ from $\pi_t$ for each prompt $x^i$ in $\mathcal{P}$
4:      Compute the rewards $r_1^i$ and $r_2^i$ for $(x^i, y_1^i), (x^i, y_2^i) \in \mathcal{D}_t$
5:      For each prompt $x^i$, determine the winning response $y_w^i$ and the losing response $y_l^i$ based on
     their rewards $r_1$ and $r_2$ and construct the training set $\mathcal{D}_t = \{(x^i, y_w^i), (x^i, y_l^i) \mid x^i \in \mathcal{P}\}$
6:      Compute the DPO loss and update the policy parameters $\pi_t$ to obtain $\pi_{t+1}$
7: **end for**

---

In Line 5 of Algorithm 2, the scalar reward values from the reward model (RM) are reduced to binary labels to determine the chosen (positive) and rejected (negative) responses. This quantization fails to leverage the full potential of the reward signals $r_1^i$ and $r_2^i$ and leads to information loss. For example, a larger reward gap indicates that there are more significant differences between the two responses that can be used to improve alignment. In contrast, DPO loss with binary labels treats all pairs equally and may lead to an inefficient training process. We hypothesize that to address these issues, it is crucial to integrate the reward scalars into the learning process more directly, ensuring that the updates to $\pi_t$ reflect both the direction and magnitude of human preferences, thus enhancing the overall alignment of the policy with desired outcomes.

To this end, we introduce a weighted DPO Loss (wDPO) that incorporates explicit reward signals directly into the loss function for online DPO training. This modification aims to enhance the training efficiency by making full use of the available reward information and better aligning the policy updates with the underlying human preferences. The wDPO Loss is derived by modifying the original DPO loss to include a weighting factor that represents the explicit rewards:

$$\mathcal{L}_{\text{wDPO}} = -\mathbb{E}_{(x,y_w,y_l)\sim\mathcal{D}} \left[ R(x, y_w, y_l) \cdot \log\left( I(x, y_w, y_l) \right) \right],$$

$$\text{where} \quad I(x, y_w, y_l) = \sigma \left( \beta_1 \log \frac{\pi_{t+1}(y_w|x)}{\pi_t(y_w|x)} - \beta_1 \log \frac{\pi_{t+1}(y_l|x)}{\pi_t(y_l|x)} \right), \quad (6)$$

$$R(x, y_w, y_l) = \sigma \left[ \beta_2 \left( r(x, y_w) - r(x, y_l) \right) \right].$$

where $I(x, y_w, y_l)$ denotes the implicit reward; $R(x, y_w, y_l)$ captures the relative preference between the winning and losing responses based on their explicit reward difference, scaled by $\beta_2$.

By incorporating these explicit rewards, wDPO improves the efficiency of the training process by prioritizing learning from pairs that show a significant difference in rewards. This approach makes the model more sensitive to examples where the distinction between preferred and less preferred responses is clear, helping it learn the essential features that distinguish highly preferred responses from those less preferred. As a result, wDPO guides policy updates more effectively toward the desired behavior, enhancing the overall training efficiency and effectiveness. This structured approach allows wDPO to leverage the full spectrum of reward information, ensuring that each training example contributes optimally to learning based on the strength of its preference signal.

## 4 EXPERIMENT

### 4.1 EXPERIMENT SETUP

Our experiments are run on 8 NVIDIA A100 80GB GPUs and the implementations are based on Huggingface TRL (von Werra et al., 2020). Similar to other online RLHF algorithms (Schulman et al., 2017; Ouyang et al., 2022a), our OPTUNE will distill the human preferences into the reward models first. On the policy training, it begins with a supervised-finetuned (SFT) model, with the carefully designed OPTUNE loss and reward-based sampling strategy for selected generations, *i.e.*, re-generating the low-score samples while reusing high-score samples.

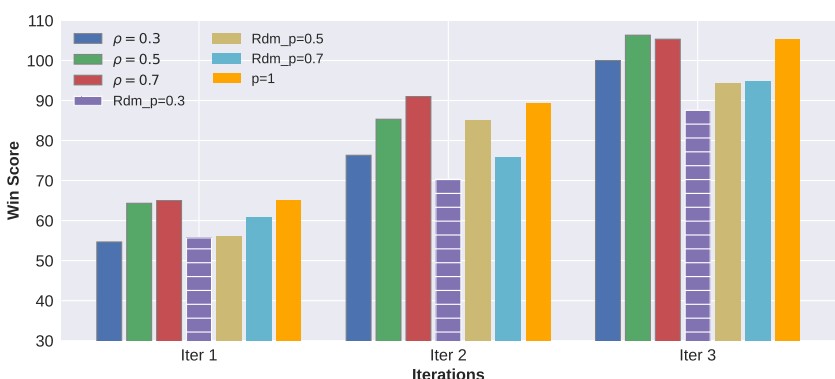

Figure 3: OPTUNE (wDPO loss): Y-axis denotes the win score against `Zephyr-7B-beta` model. Rdm_$\rho$: random selection ratio (all striped bars). Under the same selection ratio, OPTUNE'd models could perform better than the models tuned with random-selection strategy. The policies in prompt selection $\rho = 0.5$ and $\rho = 0.7$ could be comparable with the policies in $\rho = 1$ while enjoying 30% to 50% generation efficiency, which proves the effectiveness of OPTUNE.

**Dataset.** We use Ultrachat (Cui et al., 2023), which contains 200k prompts, as the preference dataset and is widely used (Chen et al., 2024c; Wu et al., 2024). Considering the budget, we only randomly sample 48k prompts on the original set to construct our prompt set which are fixed in our experiments and used as the inputs of the on-the-fly generations for iterative training of the policy.

**Models & Training.** `Zephyr-7b-sft-full` (Tunstall et al., 2023), which is SFT-ed on Ul-traChat200k dataset with decent instruction-following capability, is employed as the RL finetuning start point. For the reward models, we select the one fine-tuned by Xiong et al. (2024), which shares the same backbone, `Mistral-7B`, with the $\pi_{\text{SFT}}$ and top-ranked on RewardBench (Lambert et al., 2024). Thus, we believe it is a strong reward model that could provide informative reward signals. The prompt and generation length are both set to 512. We defer the other hyperparameters, *e.g.*, learning rate, into Appendix C.

**Baselines.** We have three baselines: (1). `Zephyr-7B-beta`, which conducts offline DPO training on the total 200k (prompt, preferred response, rejected response) triplet in UltraFeedback dataset, in which the responses come from many competitive models, e.g., GPT-3.5-turbo and GPT4. We use it as the offline baseline and expect our models under online settings could be significantly better than this baseline though we employ much less prompts for training. (2). Models tuned with selection ratio $\rho = 1.0$ and wDPO/DPO'ed for three iterations on the whole prompt set, which is under a fully online setting and has the largest generation cost. We expect the OPTUNE with smaller ratios could be on par with it. (3). Models tuned with random selection ratio. Models with OPTUNE should surpass them. We also keep the iteration 0 the same for all the OPTUNE models for fair comparison, *i.e.*, we will do one online iteration first under $\rho = 1$ and save the checkpoint & responses for further OPTUNE.

**Free-form Instruction Evaluation.** We mainly focus on free-form generation. Drawing on recent advancements (Li et al., 2023b; Zheng et al., 2023; Chiang et al., 2023) we rely on strong LLMs, *i.e.*, GPT-4 (OpenAI et al., 2023) as our judge. The LIMA test set (Zhou et al., 2023), consisting of 300 prompts, is chosen as our test set. The same rating prompt as Chen et al. (2024a) is employed to compare the responses generated by the policy with those produced by the baseline, *i.e.*, `Zephyr-7B-beta`. To counteract the positional bias identified in GPT-4's ratings (Wang et al., 2023), we collect two sets of ratings by swapping the order of test and baseline model responses. A response is deemed winning if it achieves at least one win and no more than one tie. We assess performance using the "win score", which is defined as:

$$\text{Win Score} = 50 + 100 \times \frac{n_{\text{win}} - n_{\text{lose}}}{n}, \tag{7}$$

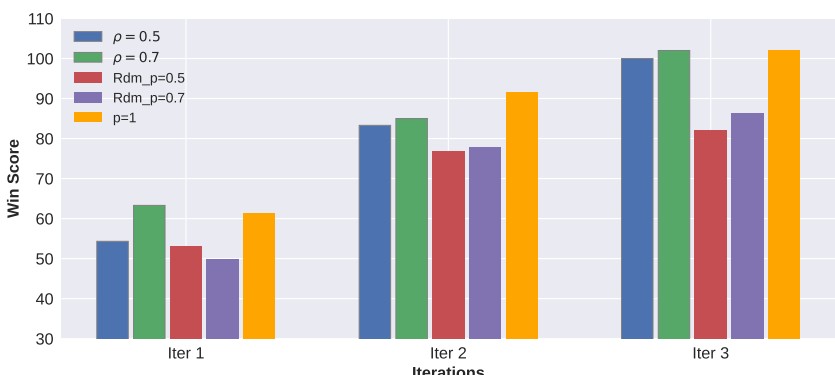

Figure 4: OPTUNE (DPO loss): Even in the special case, *i.e.*, DPO loss is a special case of our proposed wDPO, we could still have the conclusion that OPTUNE with $\rho = 0.7$ could maintain the performance but save 30% generation cost. Rdm_$\rho$: random selection ratio.

where $n_{\text{win}}$ and $n_{\text{lose}}$ are the number of examples rated as better and worse than the baseline, respectively; $n$ is the total number of evaluation examples. A Win Score $\geq 50$ indicates that the test model performs at least as well as the baseline.

**Benchmarks.**  Following LM-Evaluation-Harness (Gao et al., 2023), we test the trained policy $\pi$ on TruthfulQA (Lin et al., 2021), MMLU (Hendrycks et al., 2020a), GSM8K (Cobbe et al., 2021b), and Hellaswag (Zellers et al., 2019) to evaluate the model's ability on truthfulness, challenging multi-task solving, grade-school-level math, and common-sense reasoning. For the few-shot demo setting, we adopt the default settings in the lm-evaluation-harness and we summarize it together with the metrics in Table 5. We expect the model could also improve its performance on benchmarks since RLHF can also help the reasoning (AI@Meta, 2024; Chen et al., 2024c).

## 4.2 RESULTS ON GENERATION EFFICIENCY

**OPTune on wDPO loss.**  We first study OPTUNE on wDPO loss. We sweep $\rho = \{0.3, 0.5, 0.7, 1.0\}$ for both OPTUNE and random selection ratio and train three epochs using the same hyperparameters, *e.g.*, $\beta_2$, learning rate, etc. We defer the details of the hyperparameters into Appendix C.

In Fig. 3 we show that OPTUNE significantly outperforms the random-selection baselines and is comparable with models trained under fully online settings $\rho = 1$ while achieving 30-50% generation efficiency. We also observe an expected trend that when the number of online samples is increased, *i.e.*, larger $\rho$, the win score goes up, corroborating observations in Tang et al. (2024).

To elucidate the training efficiency of our OPTUNE further, we visualize the win score of different OPTUNE ratios with training time in Fig. 5. The training time includes generation, rewarding, and wDPO training time and we consider their sum total to provide a clear picture as to the level of efficiency OPTUNE achieves. We note that, when calculated in terms of GPU hours, the savings are 8x larger since we run the experiments on 8xA100 GPUs at a time.

**OPTune on DPO loss.**  We also verify the effectiveness of OPTUNE's selection criteria when training with the regular DPO objective (Pi et al., 2024; Yuan et al., 2024). We observe similar results on the standard DPO loss and showcase them in Fig. 4. OPTUNE matches or surpasses the performance of vanilla online DPO ($\rho = 1$) in iteration 1 and 3, though in iteration 2, it lags slightly behind the vanilla setting. However, it still enjoys 1.27-1.56x training speedup, saving 30% to 50% on generation time. Moreover, we note that OPTUNE consistently outperforms the random selection criteria across different ratios.

## 4.3 RESULTS ON TRAINING EFFICIENCY

We compare two different losses, *i.e.*, DPO and wDPO losses under different prompt selection ratios and show the results in Fig. 6. We find that wDPO with OPTUNE significantly surpasses DPO with

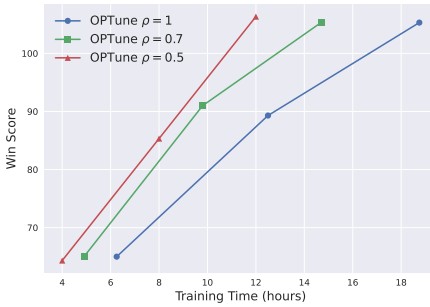
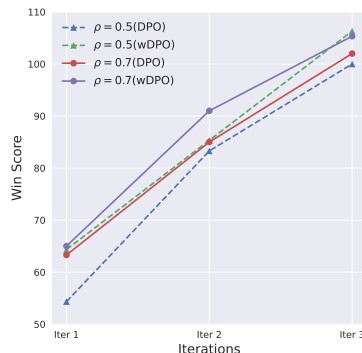

Figure 5: The win score vs. training time on different prompt selection ratios. By re-generating the responses on only half of the prompts, OPTUNE could achieve the win score on par with the vanilla online version ($\rho = 1$).

Figure 6: The online DPO vs. online wDPO under different prompt selection ratios. The dashed line denotes the OPTUNE ratio $\rho = 0.5$.

OPTUNE. We keep the training configs, *e.g.*, the learning rates in each iteration, optimizer, and the max length of the prompt & generation, exactly the same for the online wDPO and online DPO under the same ratio $\rho$. Thus, we believe the training time is almost the same for wDPO and DPO under the same ratio $\rho$. Our wDPO loss could achieve faster convergence than DPO loss, *i.e.*, it reaches the same "win score" faster than DPO does., which reflects the superiority of our proposed wDPO loss.

### 4.4 EVALUATION RESULTS ON ALPACAEVAL, BENCHMARKS, HUMAN STUDY

We provide more evaluation results including AlpacaEval (Li et al., 2023b), Benchmarks, and human studies to further test the performance of the policies trained by OPTUNE and verify the effectiveness of our method in this subsection.

Table 2: Alpaca-eval scores on the iteration-3 models trained under different settings. LC_win_rate: length-controlled win rate, which is the standard metric in `AlpacaEval-2.0`. (r): policies trained with a random selection strategy. $\rho = 0.7$ performs the best and even better than the $\rho = 1.0$.

| Model | Zephyr-7B-Beta | $\rho$=1.0 | $\rho = 0.5$(r) | $\rho$=0.5 | $\rho = 0.7$ | $\rho = 0.7$(r) |
|---|---|---|---|---|---|---|
| LC_win_rate | 13.2 | 15.43 | 15.28 | 15.63 | 16.45 | 15.39 |

**AlpacaEval.** To alleviate the concerns of evaluating the open-ended generation only on LIMA test set, we also test our trained policies on the AlpacaEval (Li et al., 2023b), which contains 805 prompts and is more diverse. Due to the limited GPT-4 API budget, we only test the models trained with wDPO loss in the final iteration (iter3). We show the results in Table 2. It aligns with the results in Fig. 4 and Fig. 3: OPTUNE is better than the random selection strategy and no selection ($\rho = 1.0$).

Table 3: Benchmark results for different prompt selection ratios. We use bold font to mark the highest score.

| Models | Hellaswag | MMLU | TruthfulQA | GSM8k | Average |
|---|---|---|---|---|---|
| Zephyr-7B-SFT | 78.54 | 55.67 | 40.37 | 32.75 | 51.83 |
| Zephyr-7B-Beta | 82.05 | 58.13 | 50.1 | 36.24 | 56.63 |
| Iter3 ($\rho$=1.0) | 81.44 | 58.49 | 45.04 | 42.3 | 56.82 |
| Iter3 (rdm=0.5) | **83.06** | 58.39 | 45.77 | 42.00 | 57.31 |
| Iter3 (rdm=0.7) | 82.17 | 58.55 | 46.22 | 42.15 | 57.27 |
| Iter3 ($\rho$=0.5) | 82.48 | **58.62** | 46.64 | 39.88 | 56.91 |
| Iter3 ($\rho$=0.7) | 82.78 | 58.46 | **46.81** | **42.53** | **57.65** |

**Benchmark Results.** The benchmark results of the trained policies are shown in Table 3 and higher values indicate better performance. The policies trained with the prompt selection ratio $\rho = 0.7$ show superiority against the offline policies (`Zephyr-7B-Beta`) and vanilla online ($\rho = 1.0$) policies regarding on the "Average" score. It also achieves the highest scores on TruthfulQA and GSM8k, showing gains in math problem-solving and factuality.

**Human Study.** To further evaluate how OPTune performs against full generation as well as random selection, we randomly select 50 responses generated by OPTune and compare them first to random selection and then to full generation ($\rho = 1.0$). On the 100 response pairs, we collect 400 ratings from 8 participants and find that participants prefer OPTune responses 24.07% of the time against 14.81% for random selection, and perform similarly to full generation with users ranking its output better 23.44% of the time, full generation 25.0% of the time and considering outputs similar 51.56 % of the time. The details of how we conduct human studies could be referred to Appendix D.

## 5 RELATED WORK

**RLHF algorithms** Proximal Policy Optimization(PPO) (Ouyang et al., 2022b; Schulman et al., 2017) is the most widely-used online preference tuning framework in the industry, which leads to the success of the ChatGPT (OpenAI et al., 2023), Gemini (Team et al., 2023), and LLaMA (Touvron et al., 2023). It requires training a reward model as a proxy of the human preference and on-the-fly generations in the online training procedure. Online DPO/wDPO stays relevant with it but the difference is that the online generation and policy updates are less frequent than PPO, in which the policy will be updated per batch. On the other hand, several offline RLHF methods such as DPO (Rafailov et al., 2023), IPO (Azar et al., 2024a), KTO (Ethayarajh et al., 2024), and SLIC-HF (Zhao et al., 2023) also show promises for learning of human preference. These methods are considered offline because their preference datasets are kept unchanged during RLHF but the performance of the offline RLHF could not be on par with the online version (Tang et al., 2024; Dong et al., 2024). Thus, in this work, we focus on investigating online iterative RLHF, which demands substantial computational resources for on-the-fly sampling from the policies. OPTUNE is proposed to reduce the cost in the regeneration process by selecting a subset of prompts to regenerate while keeping the outstanding performance of the trained models.

**Prompt Selection.** A powerful LLM usually requires high-quality training data, and the community has focused on creating high-quality instruction finetuning (IF) datasets, either via distilling of the SOTA API LLMs (Taori et al., 2023; Peng et al., 2023; Chiang et al., 2023) or requiring experienced human annotators (Conover et al., 2023; Ouyang et al., 2022a). But there are still low-quality examples in these IF datasets and a series of data selection strategies (Chen et al., 2023b; Li et al., 2023a; Cao et al., 2024) are proposed to further enhance the quality of datasets by filtering out these data, which shares the same objective with the OPTUNE: optimizing towards the training data quality. However, these data selection approaches are not ideal for prompt selection in the iterative RLHF paradigm as they primarily focus on the quality of the responses, not targeting selecting the prompt for efficient data exploration.

**Inference Speedup of LLMs.** One orthogonal direction to our method is the inference speedup of LLMs. Traditionally, batch inference and Key-Value (KV) cache (Ge et al., 2023) are employed to accelerate the decoding process, but they consume substantial GPU memory and hinder the utilization of large batch sizes. Thus, some works (Shazeer, 2019; Ainslie et al., 2023; Xiao et al., 2023; Dettmers et al., 2022) are proposed to reduce the memory used by KV cache through changing model architecture or using quantization techniques. On the other hand, some other approaches (Leviathan et al., 2023; Chen et al., 2023a; Cai et al., 2024) are proposed to minimize the number of decoding steps to speed up the inference of LLMs. Compared to it, OPTUNE achieves efficiency by reusing the generations in the previous step. But all these inference speedup techniques can be used for the selected prompts of OPTUNE, providing further faster generation speed.

**Evaluation of LLMs.** To evaluate the instruction-following ability of the policies in iterative RLHF procedure, we employ GPT-4 (OpenAI et al., 2023) as our judge and employ LIMA (Zhou et al., 2024) test set which contains 300 prompts and larger than the MT-bench (Zheng et al., 2023) (80 prompts), Koala (Geng et al., 2023) (180 prompts), and WizardLM test set (Xu et al., 2023) (218 prompts). AlpacaEval (Li et al., 2023b) is also employed to evaluate the trained policy on instruction-following ability more comprehensively. Moreover, following the previous works (Chen et al., 2023b; 2024b), we also include human study for a side-by-side comparison of the model responses and test the models on four most commonly used benchmarks, TruthfulQA (Lin et al., 2021), MMLU (Hendrycks et al., 2020b), GSM8K (Cobbe et al., 2021b), and Hellaswag (Zellers et al., 2019).

## 6 DISCUSSIONS & CONCLUSION

To sum up, we introduced OPTUNE in this work, a novel approach to enhance the training and generation efficiency of online RLHF by selectively regenerating only the lowest-reward responses and representing the reward gap explicitly in our wDPO objective. This method focuses computational resources on the most informative samples, significantly reducing the need for full-scale data regeneration and achieving up to 2x in generation efficiency and a 1.56x speedup in training efficiency. Our comprehensive experiments show that OPTUNE maintains or improves the alignment of LLMs with human preferences. Finally, we believe OPTUNE could also be applied to other online RLHF algorithms such as Best-of-N (Stiennon et al., 2020) and PPO (Schulman et al., 2017), since PPO has a replay buffer which contains "off-policy" examples and we could select the prompts using the same strategy to encourage the generations on the low-reward prompts, which we leave for the future work.

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

## A  LIMITATIONS

Despite the advancements presented by OPTUNE in online RLHF, OPTUNE's performance heavily relies on the accuracy and consistency of the reward model (RM). If the RM does not effectively capture the nuances of human preferences or suffers from biases, the efficiency gains from our approach could lead to suboptimal policy training.

## B  BROADER IMPACT

In this paper, we introduce OPTUNE, enhancing the training efficiency and generation efficiency of the online RLHF. The broader impacts of this study are two-fold:

1. **Advancing AI Alignment with Human Values:** The proposed OPTUNE significantly improves the alignment of AI behaviors with human preferences. This enhancement is vital for deploying AI in sensitive applications, ensuring that AI responses adhere closely to human ethical standards.

2. **Enhancing Efficiency in AI Training:** OPTUNE accelerates the LLM training process without compromising output quality. This advance reduces computational bottlenecks, enabling faster development cycles and making high-performing AI models more accessible, especially to organizations with limited computational resources.

## C  HYPERPARAMETERS

We use learning rate = 5e-7 for DPO/wDPO training with RMSProp (Hinton, 2012) as our optimizer; the warmup ratio is set to 0.1 and the batch size is 128. To encourage the model's exploration, we choose `top_p`=0.9 and temperature T=1.0 as the generation config in data generation part.

## D  HUMAN STUDY

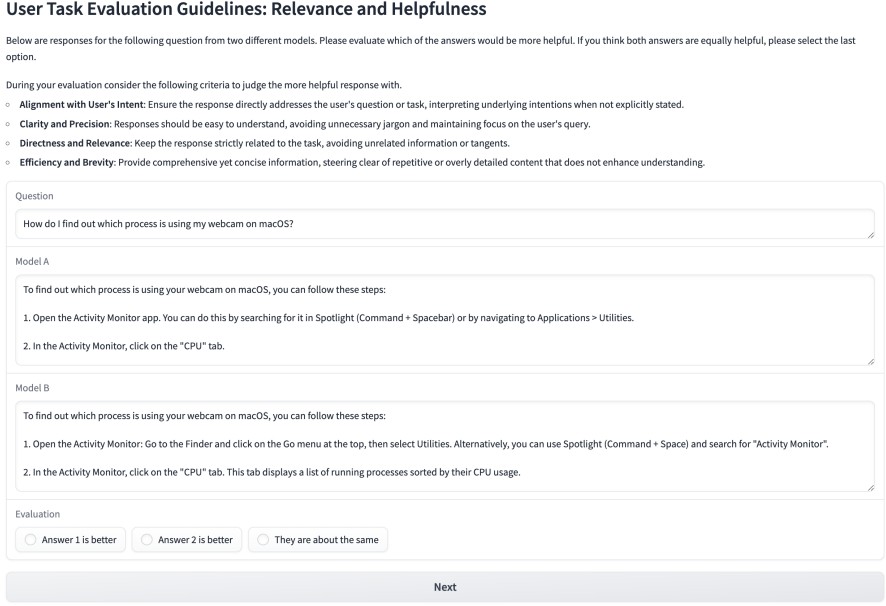

Figure 7: UI for the human study. At each step, the participants are presented with the prompt and generations from two models and asked to indicate their preferences.

For human study, we randomly choose 50 prompts from the original LIMA test set and present them to the participants. We recruit 8 volunteer students as the participants in the human study. For each

prompt, we create two comparison pairs, one pair contains responses from the two policies trained by OPTUNE $\rho = 0.7$ and OPTUNE $\rho = 1.0$, respectively; another pair contains responses from the policy trained by OPTUNE $\rho = 0.7$ and the policy trained by random selection $\rho = 0.7$. Through this way, we create a total of 100 unique pairs. Each participant is presented with 50 randomly selected pairs from these 100 unique pairs and is asked to choose which one they prefer with the guiding criteria based on (Chen et al., 2024b). In the UI interface, they can indicate a preference or a tie as shown in Fig. 7.

Overall we obtain 400 ratings, 200 for each comparison. The distribution is shown in Table 4.

We also provide the user guidelines which is used in our human study:

> Below are responses to the following questions from two different models. Please evaluate which of the answers would be more helpful. If you think both answers are equally helpful, please select the last option.
>
> During your evaluation, consider the following criteria to judge the more helpful response:
>
> - **Alignment with User's Intent**: Ensure the response directly addresses the user's question or task, interpreting underlying intentions when not explicitly stated.
> - **Clarity and Precision**: Responses should be easy to understand, avoiding unnecessary jargon and maintaining focus on the user's query.
> - **Directness and Relevance**: Keep the response strictly related to the task, avoiding unrelated information or tangents.
> - **Efficiency and Brevity**: Provide comprehensive yet concise information, steering clear of repetitive or overly detailed content that does not enhance understanding.

| Comparison | OPTune Win (%) | Loss (%) | Tie (%) |
|---|---|---|---|
| OPTUNE $\rho = 0.7$ vs OPTUNE $\rho = 1.0$ | 23.44 | 25.00 | 51.56 |
| OPTUNE $\rho = 0.7$ vs Rdm $\rho = 0.7$ | 24.07 | 14.81 | 61.11 |

Table 4: Results of the human study, the pairs of responses to each prompt are rated by 4 people on average.

## E  THE BENCHMARK SETTINGS

Table 5: The metrics and few-shot demos for each benchmark. It is the standard setting in LM-Harness-Evaluation repo (Gao et al., 2023)

| Datasets | TruthfulQA | GSM8k | HellaSwag | MMLU |
|---|---|---|---|---|
| # few-shot | 0 | 5 | 0 | 0 |
| Metric | mc2 | acc | acc_norm | acc |

## F  RATING PROMPT

Following Chen et al. (2024a), we also use the GPT-4 rating prompt in the original Vicuna blog post [1] and we provide the detailed form in Table 6.

---

[1] https://lmsys.org/blog/2023-03-30-vicuna/

Table 6: The GPT4 evaluation prompt.

[System Prompt]
You are a helpful and precise assistant for checking the quality of the answers.
[User Prompt]
[Question]
[The Start of Assistant1's Answer]
Answer 1
[The End of Assistant1's Answer]
[The Start of Assistant2's Answer]
Answer 2
[The End of Assistant2's Answer]

We would like to request your feedback on the performance of two AI assistants in response to the user question displayed above. Please rate the helpfulness, relevance, accuracy, and level of details of their responses. Each assistant receives an overall score on a scale of 1 to 10, where a higher score indicates better overall performance. Please first output a single line containing only two values indicating the scores for Assistant 1 and 2, respectively. The two scores are separated by a space. In the subsequent line, please provide a comprehensive explanation of your evaluation, avoiding any potential bias and ensuring that the order in which the responses were presented does not affect your judgment.

