# OpenReview forum: "OPTune: Efficient Online Preference Tuning"
_ICLR.cc/2025/Conference — ICLR 2025 Conference Withdrawn Submission_

### Official Review · Reviewer_mgRJ · 2024-10-31

**Soundness:** 2
**Presentation:** 1
**Contribution:** 2
**Rating:** 3
**Confidence:** 3

**Summary:**

The paper proposes two improvement for online preference learning: one is reward-based prompt selection, and the other one is weighted DPO. For reward-based prompt selection, the paper proposes that between each iteration of online DPO, one should only regenerate certain proportion of the reponses to the prompts that have the lowest rewards. For weighted DPO, the paper proposes to add a weight term in the DPO loss based on the reward difference in each pair of generation. The paper performs experiments to show that, with  reward-based prompt selection, for both DPO and wDPO loss, selecting the right portion of regeneration will improve the efficiency of online DPO without sacraficing the performance.

**Strengths:**

1. The experiment is performed on a 7B model and the result suggests that with a right ratio for regeneration, the proposed method indeed improves training efficiency without decreasing the performance.
2. The experiment has reasonable comparison with random subselection and shows that random subselection does not work as well as the proposed method.

**Weaknesses:**

1. It is unclear to me that ranking the prompts by absolute reward makes sense, especially if the reward model is trained by BT loss. For each fixed prompt, the BT loss only cares about the difference between two responses, so difference prompts may induce a difference biased of the corresponding completion. Thus having a low reward does not necessarily mean that the model is currently performing bad on the prompt. Honestly I might be describing the procedure wrong because I don't see a clear definition of "ranking prompt by reward" unless I am missing something.

2. Also it is unclear to me why the wDPO loss makes sense. If the reward gap between two generations are large, it is likely that the pair is already easy for the model, and the term might not even contribute to the training - why not doing the inverse weight?

3. There is no information how table 1 is generated, and it seems like it is the major motivation for the proposed method. More details should be provided, especially to show that all procedures are fully optimized - for example, for generation it seems that using vllm to speed up the inference is the common approach.

4. Frankly, the paper lacks basic rigors.
- In section 3.1, the important concept of "reward gain" is not defined so the motivation part is very confusing.
- In line 7 of alg 1, the prompt $x^i$ is already popped, then from line 12 should we never see the recently added pairs in $\mathcal{R}_t$?
- In line 21 of alg 1, how is the ranking computed?
- nits: a) in eq (2), the two terms inside KL are not distribution. 2) eq (2) uses $\alpha$ and the following uses $\beta$. 3) in line 167 $\mathcal{P}$ is not defined.

**Questions:**

see above

---

### Official Review · Reviewer_8QcR · 2024-11-03

**Soundness:** 1
**Presentation:** 2
**Contribution:** 1
**Rating:** 3
**Confidence:** 5

**Summary:**

The paper addresses the high-cost issue of online alignment. It proposes a method aimed at improving the efficiency of online  alignment, specifically consisting of two parts. First, only the lowest-rewarded responses generated under the latest LLM policy are regenerated and updated. Second, the loss function is modified to assign higher weights to response pairs that contribute more during training.
Similar methods to the two improvements proposed in this paper have already emerged within the community. Further, the experimental section lacks a proper evaluation of the improvements due to the choice of an outdated and subpar baseline Zephyr 7B Beta (Alpaca eval rank 131).

**Strengths:**

The writing is relatively clear.

**Weaknesses:**

1. Lack of Innovation
Over the past year, the alignment community has proposed numerous methods similar to those used in this paper. As early as the Llama 2 Technical Report, the approach of directly incorporating the score difference between two responses into the loss function was introduced. Although the Llama 2 Technical Report is cited in the Related Work section, there is no comparative discussion with Llama 2 or other similar works in Section 3.2.
2. Incomplete Experiments and Lack of Analysis
This paper introduces a scaling factor, beta 2, to amplify the score difference and combines it with the original DPO loss function via multiplication rather than addition. However, the motivation behind this approach is not explained. More importantly, there is no ablation study to compare the impact of different values for the scaling factor beta 2 or other ways of incorporating score differences.
3. Unconvincing Choice of Baseline
The entire experimental section only includes the Zephyr model as a baseline, with no comparisons to other baselines.
Currently,  on the alpaca_eval board(https://tatsu-lab.github.io/alpaca_eval/) Zephyr-7B-Beta has a win rate of 13.2%, ranking 131st. When controlling for model size and listing only models with 8B or fewer parameters, there are other fine-tuned models based on comparable foundation models. These include the Gemma series (e.g., Gemma-2-9B-it-SimPO, rank 8; Gemma-2-9B-it-DPO, rank 10), Llama3-based fine-tuned models like Llama-3-Instruct-8B-WPO-HB-v2 (rank 20), and Mistral 7B-based models like Storm-7B (rank 24). Without comparisons to any of these other similarly sized fine-tuned models, the paper’s conclusions are difficult to accept.

Longitudinally, the Zephyr model has several later versions, including FsfairX-Zephyr-Chat-v0.1 (rank 50, LC win rate 34.8%), ExPO + Zephyr 7B Beta (rank 128, LC win rate 14.0%), and Zephyr 7B Beta (rank 131, LC win rate 13.2%). The paper only selected Zephyr 7B Beta, which ranks last among these, as its baseline, with a win rate only 40% of the current best Zephyr model. Additionally, instead of using the common win rate metric, the paper employs win score, making it difficult to directly compare the performance of Optune against existing models.

**Questions:**

Why didn’t the authors choose a stronger model from the Zephyr series as a baseline, or conduct comparisons with other models like Llama-3-Instruct-8B-WPO-HB-v2?

---

### Official Review · Reviewer_Vdwc · 2024-11-03

**Soundness:** 3
**Presentation:** 3
**Contribution:** 2
**Rating:** 5
**Confidence:** 4

**Summary:**

The authors propose a more generation-efficient procedure for online RLHF by preferentially sampling responses from prompts that had low rewards and weighting samples by the reward gap in the online DPO loss.

**Strengths:**

- Show across multiple experiments that the proposed strategy outperforms a random selection strategy.

**Weaknesses:**

- Lack of relevant baselines on sample selection: a pretty common strategy in RLHF is to pick prompts that had the largest "margin" between the winner and the loser for further training (e.g. https://arxiv.org/abs/2404.03715). Could you compare your strategy against this technique?

- Lack of relevant baselines on policy optimization: a variety of papers have already noted that IPO / DPO ignore the gap in reward between the winning and losing completions. Could you compare against at least one of these (e.g. REBEL: https://arxiv.org/abs/2404.16767).

**Questions:**

1. Would it be possible to provide some sort of conceptual grounding for your proposed prompt selection strategy? I could imagine a connection to the pessimism principle in RL.

---

### Official Review · Reviewer_t2sx · 2024-11-03

**Soundness:** 3
**Presentation:** 2
**Contribution:** 2
**Rating:** 5
**Confidence:** 3

**Summary:**

This paper propose OPTune, an approach to enhance the both generation and training efficiency of online preference tuning for LLMs alignment.
To improve the generation efficiency, OPTune selects prompts whose regenerated responses are likely to provide more informative and higher-quality training signals.
In addition, weighted DPO is proposed to improve the training efficiency by modelling the reward gap of response pairs.
Empirical results show that LLMs tuned with OPTune maintain instruction-following benefits and achieve faster training speeds compared to standard preference tuning methods.

**Strengths:**

The paper is well-written and well-orginized.

Considering online DPO takes more time than the original offline method, improving its efficiency is of great significance.

**Weaknesses:**

Given that iterative DPO often utilizes different prompts in different iterations [1] for avoid overfitting or overoptimization [2],  it is not clear how the proposed method can be used in such scenarios.

The performance of the models corresponding to different selection ratios in Table 2 is not very different and is generally low, which cannot explain the effectiveness of the method.


References:

[1] Meng Y, Xia M, Chen D. Simpo: Simple preference optimization with a reference-free reward[C]. NeurIPS, 2024.

[2] Rafailov R, Chittepu Y, Park R, et al. Scaling laws for reward model overoptimization in direct alignment algorithms[J]. arXiv preprint arXiv:2406.02900, 2024.

**Questions:**

How to rank the response pairs? Do you use the average rewards of preferred and less preferred responses? Is there a better prompt selection method suitable for wDPO?

Regarding the experimental configuration in Table 1: How many responses were generated for each prompt?

Did the author observe overoptimization reusing the same prompts for each iterations?

---

### Official Review · Reviewer_NRtG · 2024-11-04

**Soundness:** 3
**Presentation:** 3
**Contribution:** 2
**Rating:** 5
**Confidence:** 4

**Summary:**

This paper targets LLM alignment with human preferences in an online manner. OPTune involves two main strategies to reduce computational costs while maintaining alignment quality, including selective generation and weighted DPO loss. The authors conduct experiments using OPTUNE with LLMs and report a 1.27–1.56x speedup in training while maintaining or even improving model alignment.

**Strengths:**

1. OPTune achieves notable computational savings in data generation and training, reducing costs for online RLHF while preserving alignment quality.
2. By focusing on low-reward prompts, OPTune avoids unnecessary regeneration, which is a pragmatic approach to improve efficiency.
3. Using weighted DPO loss changes binary signals to dense signals, improving improving alignment through prioritizing high-utility samples.

**Weaknesses:**

1. The choice of the ratio of re-generated prompts $\rho$ can be a key factor of OPTune. Though the authors conduct experiments with different $\rho$s, the authors do not provide direct insights on how to choose $\rho$ to balance between efficiency and performance.
2. Online DPO (without weighted loss) should be the most related baseline for this paper. Though some experiments are conducted, the authors do not sufficiently evaluate OPTune's superiority over online DPO.
3. In Table 3, the performance in TrustfulQA is incorrectly bold. The offline DPO model has higher performance.
4. The choice of $\beta_2$ in the weighted loss is significant while the authors do not reveal any insight or related experiments on it.

**Questions:**

1. A choice of small  $\rho$ can speed up the training process. However, we may improve the efficiency by directly reduce the training epochs while enlarging the learning rate, which may bring more significant speedups. Will an online training method with dynamic learning rate adjustment have better efficiency?
2. Samples whose reward gap between positive and negative responses is high may dominate the learning loss. Does the training curve show more significant instability than DPO?
3. Will the un-re-generated responses be over-optimized?

---

### Note · Authors · 2024-11-25

I have read and agree with the venue's withdrawal policy on behalf of myself and my co-authors.